# A Cross-Sectional Study on Physical Activity and Burnout among Family Physicians in Slovenia during the First Year of the COVID-19 Pandemic: Are the Results Alarming Enough to Convince Decision-Makers to Support Family Medicine?

**DOI:** 10.3390/healthcare12010028

**Published:** 2023-12-22

**Authors:** Ksenija Tušek Bunc, Janja Uplaznik, Polona Selič-Zupančič

**Affiliations:** 1Dr. Adolf Drolc Health Center Maribor, 2000 Maribor, Slovenia; 2Department of Family Medicine, Faculty of Medicine, University of Maribor, 2000 Maribor, Slovenia; 3Department of Family Medicine, Faculty of Medicine, University of Ljubljana, 1000 Ljubljana, Slovenia; 4Department of Psychology, Faculty of Medicine, University of Maribor, 2000 Maribor, Slovenia

**Keywords:** physical activity, burnout, primary care, family physician, family medicine trainee, COVID-19 pandemic

## Abstract

Physical activity has a positive effect on general health, but its influence on burnout remains unclear. The aim of this study was to determine the association between physical activity and the incidence of burnout in Slovenian family physicians (FPs) and family medicine trainees (FMTs) during the COVID-19 pandemic, which exacerbated the already-existing problem of burnout. We conducted a cross-sectional observational study among Slovenian family physicians and FMTs in which sociodemographic variables, the type and duration of physical activity, and an assessment of burnout were collected using the Maslach Burnout Inventory. Comparisons between groups were made using the independent-samples *t*-test, Fisher’s exact test, and the Wilcoxon sign-rank test. A value of *p* < 0.05 determined the limit of statistical significance. Of 1230 FPs and FMTs invited to participate, 282 completed the survey (22.9% response rate); there were 243 (86.2%) FPs and 39 (13.8%) FMTs. The overall rating for burnout during the pandemic was high, at 48.6% of FPs and FMTs; 62.8% of respondents reported a high rating for emotional exhaustion and 40.1% for depersonalization. Compared to FMTs, emotional exhaustion and total burnout scores were higher for FPs (*p* < 0.001 and *p* = 0.010, respectively), but work status was not related to personal acomplishment, which 53.5% of all participants rated as low. Physical activity did not appear to be a statistically significant factor in the occurrence of burnout during the pandemic. Therefore, work status or occupational role (FP vs. FMT) should be thoroughly investigated in the future along with some other factors and a better response rate.

## 1. Introduction

### 1.1. Burnout in Health Professions

Burnout is a work-related psychological syndrome that is often a response to prolonged exposure to emotional and interpersonal stressors in the workplace and manifests as negative self-esteem and attitudes toward work [1,2]. It has been described as a combination of emotional exhaustion (EE) and depersonalization (D) along with low personal accomplishment (PA) [3]. Professions that deal with people are most at risk, such as health professionals, psychotherapists, teachers, police officers, etc. In these professions, chronic and excessive demands can lead to EE or D, which gradually affects the sense of self-efficacy. Among professions that help people, physicians are particularly vulnerable to burnout because of the nature of their work. There have been numerous studies of burnout among physicians in various specialties [4,5,6] and also among general practitioners [7,8,9,10,11]. In January 2020, 42% of physicians worldwide reported symptoms of burnout, with the highest incidence in critical care, emergency medicine, and family medicine [12].

Burnout in health professions has been associated with mood disorders, substance and alcohol use, suicidal ideation, and accidents [13,14,15]. Occupational burnout has also been associated with poorer quality of patient care and lower patient satisfaction [16,17].

### 1.2. The Effect of Physical Activity on Reducing Burnout Symptoms

Physical activity, as any movement of the body that involves skeletal muscles and results in energy expenditure, is performed during leisure time, work, and transportation [18]. It plays an important role not only in protecting against chronic noncommunicable diseases but also in protecting mental and cognitive health and increasing feelings of overall satisfaction [19]. Regular physical activity is an important protective factor against chronic noncommunicable diseases, which are a major financial burden in developed and developing countries and are responsible for the loss of active life years [20]. Globally, the proportion of people who engage in regular physical activity is relatively low. In developing countries, this proportion is twice as low as in developed countries (where it is about 63%); one in three women and one in four men do not reach the recommended level of physical activity [18]. Worldwide studies also show that physicians are likely to be more physically active than the general population. During residency, physical activity among trainees decreases compared with medical school but later increases [21,22].

It has been suggested that regular physical activity may be beneficial, particularly in preventing burnout, but worldwide studies disagree on this point; some have observed less EE in physically active individuals, whereas no significant changes have been found in D and PA [23,24]. Physical activity alone does not have a statistically significant effect on reducing burnout symptoms once they occur [25], and there is a lack of high-quality intervention studies in this area that further define the importance of regular physical activity in preventing burnout in physicians [23].

### 1.3. The COVID-19 Pandemic, Physical Activity, and Its Influence on the Incidence of Burnout among Primary Care Physicians

In March 2020, the World Health Organization and the Slovenian government declared an outbreak of a global pandemic of COVID-19 (referred to as a pandemic in the remainder of this text). Even before this outbreak, burnout was recognized as a public health crisis, and there were alarmingly high rates of occupational burnout among trainees and physicians [26,27,28]. The pandemic has dramatically increased stress for health professionals because of the personal risk of illness, increasing numbers of patients with poor outcomes, few proven treatment options to prevent morbidity and mortality, and tremendous demand for healthcare resources [29,30]. In the first year of the pandemic, from April 2020 to March 2021, in the United States of America a mixed-methods analysis of the experiences of healthcare workers on the frontline and on the margins of COVID-19 was conducted. Of the more than 1200 respondents, 12.1% reported no signs of burnout, while 21.9% reported four or more signs. The qualitative analysis provided additional insights and identified professional identity, intrinsic stressors, extrinsic factors, and coping strategies as the most important themes [31]. A meta-analysis of burnout among healthcare workers during the COVID-19 pandemic revealed a prevalence of 37.4% (95% CI 14.8–67.2%) [32]. Burnout among physicians has increased, particularly among those working directly with COVID-19 patients [33,34], especially in the EE dimension. Nevertheless, no significant change in the prevalence of burnout among internists and FPs in Japan was observed throughout the COVID-19 pandemic. However, self-quarantine was associated with an exacerbation of burnout levels [35]. Trainees experienced the same stressful situations during the pandemic, but with the added concern that their training would be interrupted.

The aim of the present study was to determine the relationship between physical activity during the pandemic and its influence on the occurrence of burnout in FPs and FMTs in Slovenia. As far as we know, no data on the frequency of physical activity in FPs could be found and there was no study investigating the association between physical activity and burnout in Slovenia, especially in FPs during the pandemic. Therefore, this study is the first investigation of this interesting topic and had several objectives, namely to determine the proportion of FPs in Slovenia who were regularly physically active before and during the studied period of the pandemic, to identify factors associated with regular physical activity in FPs, to investigate the association between regular physical activity and burnout, and finally, to determine the prevalence of burnout among FPs and FMTs in Slovenia. Several hypotheses were tested: H1: Before the COVID-19 pandemic, more than 75% of Slovenian FPs were physically active according to WHO guidelines (i.e., at least 150 min of moderate or at least 75 min of high-intensity exercise per week, or a combination of both). H2: FPs in Slovenia were less regularly physically active during the pandemic than before the pandemic H3: The incidence of burnout (expressed by MBI_TOT_) was lower among FPs who were regularly physically active before the pandemic than among those who were not regularly physically active. H4: The incidence of burnout (expressed by MBI_TOT_) was lower among FPs who were regularly physically active during the pandemic than among those who were not regularly physically active. When testing our hypotheses, we also considered possible differences between FPs and FMTs to clarify the relationship between burnout, physical activity, and occupational role/work status.

## 2. Materials and Methods

### 2.1. Participants and Procedures

The cross-sectional study was conducted between May and June 2021. The population consisted of FPs and FMTs in Slovenia, who were invited to participate by email. Invitations were sent twice, two weeks apart, by the Medical Chamber of Slovenia and the Association of Family Physicians of Slovenia. All FPs who had given their consent to the General Data Protection Regulation (GDPR) were invited. At the time of data collection, there were 1230 FPs and FMTs in Slovenia. The inclusion criteria were the specialty (family medicine) and consent to the GDPR. The exclusion criterion was refusal to participate in the study within the specialty of family medicine. The data were collected via an online survey. Of all those invited, 282 completed online questionnaires were returned, of which only 39 were FMTs (an overall response rate of 22.9%; only 17.6% for FMTs).

### 2.2. Measures

The first part of the questionnaire addressed the following sociodemographic data of the participants: gender, age, place of residence, profession (physician/education), experience in family medicine, marital status, parenthood, and presence of chronic diseases.

The second part dealt with physical activity by asking about the number of minutes per week spent in moderate and vigorous physical activity, as defined by the WHO [21]. The questionnaire also assessed the time spent in moderate and vigorous physical activity before and during the pandemic (after March 2020), as well as the number of strengthening exercises performed per week. We used the WHO definition of at least 150 min of moderate or at least 75 min of high-intensity physical activity per week, or a combination of both, as a criterion for regular physical activity [21].

In our literature search, we found no precise guidelines or suggestions for combinations of the two types of physical activity. Therefore, we initially defined our own and used the following combinations in the analysis: either 1–2 h of moderate activity and 0.5–1 h of vigorous activity; or 0.5–1 h of moderate activity and 1–1.25 h of vigorous activity; or 2–2.5 h of moderate activity and up to half an hour of vigorous activity. However, looking at the WHO definition (at least 150 min of moderate or at least 75 min of high-intensity physical activity per week, or a combination of both), it can be argued that 1 min of high-intensity physical activity is equivalent to 2 min of moderate-intensity physical activity. With this in mind, we have additionally chosen a combination rule that is more consistent with the WHO definition: regular physical activity = {(1 if (minutes of moderate activity + 2 min of high activity) > 150/0 otherwise} to check our first calculation. The categories of moderate physical activity were converted into minutes as follows: less than 0.5 h per week to 15 min; 0.5–1 h per week to 45 min; 1–2 h per week to 90 min; 2–2.5 h per week to 135 min; 2.5–3.5 h per week to 180 min; 3.5–5 h per week to 255 min; and more than 5 h per week to 330 min. The categories of intense physical activity were converted into minutes as follows: less than 15 min weekly at 7.5 × 2 = 15 min; 15–30 min weekly at 22.5 × 2 = 45 min; 0.5–1 h weekly at 45 × 2 = 90 min; 1–1.5 h weekly at 75 × 2 = 150 min; 1.5–2 h weekly at 105 × 2 = 210 min; 2–2.5 h/week at 135 × 2 = 270 min; and more than 2.5 h/week at 165 × 2 = 330 min. We then added the moderate and vigorous weekly physical activity to obtain a total number of minutes. If the total number of minutes exceeded 150, the study participant was classified as regularly physically active. This mathematical approach was 100% consistent with our observational approach described above, which increases the validity of our findings.

The third part of the survey consisted of the Maslach Burnout Inventory (MBI), which divides burnout into three subcategories: EE, D, and PA [3]. EE stands for a feeling of excessive demands and exhaustion of one’s own psycho-physical resources; the person is physically and emotionally exhausted, has the feeling of having no more energy for work, the feeling of no longer being able to do it—this is the most important component of burnout. D (also called withdrawal or cynicism) is the related component of burnout; the person withdraws from work, does not care about it and their own performance, and becomes indifferent to work, colleagues, or clients. Reduced PA goes hand in hand with feeling less competent for the job, with low productivity; the person does not feel able to do their job as efficiently as they used to be able to. The standardized MBI questionnaire contains 22 statements describing feelings and attitudes related to work. Scores are measured on a seven-point numerical scale ranging from 0 (never) to 6 (every day). A higher total score for EE (9 statements) and D (5 statements) represents a higher rate of burnout on each dimension. As with PA (8 statements), a lower total score represents higher burnout. The sum of all values corresponds to the total burnout (MBI_TOT_). The EE scale ranged from 0 to 54 points (0 best, 54 worst), the D scale from 0 to 30 (0 best, 30 worst), and the PA scale from 0 to 48 (0 worst, 48 best). For the MBI_TOT_ score, PA was inverted, resulting in a scale range of 0 to 132 points (0 best, 132 worst). The internal consistency of all MBI subcategories, as measured by Cronbach’s alpha, was above 0.7 (alpha of 0.956 for EE, alpha of 0.801 for D, and alpha of 0.838 for PA).

### 2.3. Data Analysis

The results of categorical variables were presented as frequencies with the corresponding percentages, while continuous variables were represented by means and standard deviations.

Comparisons between groups were made using the independent-samples *t*-test, Fisher’s exact test, chi-square test, and Wilcoxon sign-rank test. Cohen’s d was used as an effect size to represent the magnitude of differences between the two groups for a given variable [36]. Pearson’s correlation coefficient was used to calculate the correlations between MBI subcategories. A value of *p* < 0.05 determined the limit of statistical significance.

The incidence of burnout in FPs and FMTs was determined by calculating the mean of EE 30.4 ± 12.7, D 11.1 ± 5.6, and PA 31.0 ± 6.0. The cumulative value for MBI_TOT_ was 58.5 ± 21.5.

Statistical power analysis: The lowest sample size included in a bivariate comparison was 39 FMTs. Using the independent-samples *t*-test (with a two-sided alpha of 0.05, a mean effect size of 0.5, a sample of 243 FPs, and 39 FMTs), we were able to achieve a statistical power of more than 80% [37].

## 3. Results

### 3.1. Demographic Characteristics and Prevalence and Intensity of Burnout among Slovenian FPs during the Pandemic

Of 1230 invited FPs and FMTs, 323 participated in the survey. In total, 41 (12.7%) questionnaires were incomplete and excluded from the analysis. Finally, the 282 completed questionnaires were included in further analysis, representing a response rate of 22.9%.

A total of 221 (78.4%) women and 61 (21.6%) men aged 44.8 ± 11.1 years (between 26 and 72 years) took part in the study. Only 39 (13.8%) of the participants were FMTs (aged 31.7 ± 6.2 years) and 243 (86.2%) were FPs aged 46.9 ± 10.2 years (*p* < 0.001). (Table 1).

The percentage of individuals with high burnout expressed by MBI_TOT_ was 48.6%. (Table 1). There were no statistically significant differences in reported burnout by gender, place of residence, marital status, parenthood, or presence of chronic disease.

Pearson’s correlation coefficient showed a positive correlation of medium strength between EE and D (r = 0.693, *p* < 0.001). The correlations between EE and PA and between D and PA were both negative and of medium strength (r = −0.598, *p* < 0.001 and r = −0.658, *p* < 0.001, respectively).

### 3.2. How Much Regular (Recommended) Physical Activity Did the FPs Do?

Of the participants who were regularly physically active before the pandemic, 100 (35.5%) reported engaging in moderate exercise for at least 150 min per week and 62 (22.0%) reported engaging in vigorous exercise for at least 75 min per week (Table 2). For the combination that met the criteria for regular physical activity, we set a threshold of 1–2 h of moderate activity plus 0.5–1 h of vigorous activity, or 0.5–1 h of moderate activity plus 1–1.25 h of vigorous activity, or 2–2.5 h of moderate activity and up to half an hour of vigorous activity. Only 177 (62.8%) participants reached the threshold. The χ^2^ test showed that the results were statistically significantly below the 75% expectation (χ^2^ = 23.278; *p* < 0.001). Before the COVID-19 pandemic, no more than 75% of FPs in Slovenia were physically active according to the WHO guidelines (i.e., at least 150 min of moderate or at least 75 min of vigorous exercise per week, or a combination of both).

Regular physical activity as defined by the WHO was reported by 127 (45.0%) participants; of these, 112 (46.1%) were FPs and 15 (38.5%) were FMTs (*p* = 0.392). Broken down by gender, 105 (47.5%) women and 22 (36.1%) men were regularly physically active before 2020 (*p* = 0.146). The weekly duration of moderate and vigorous physical activity is shown in Table 3. The proportion of participants who reported engaging in less than half an hour of moderate physical activity during the pandemic increased from 8.5% to 22.3%, and the proportion of regular moderate physical activity decreased statistically significantly during the pandemic (*p* < 0.001). At the same time, the proportion of FPs who reported engaging in less than fifteen minutes of vigorous physical activity during the pandemic increased from 29.4% to 44.0%, with a statistically significant decrease in the proportion of vigorous physical activity during the pandemic (*p* < 0.001).

The comparison before and during the pandemic showed that 62 (22.0%) participants gave up their regular physical activity and only 12 (4.3%) became regularly physically active (*p* < 0.001). Regarding the level of physical activity before and during COVID-19, participants reported that moderate physical activity was reduced in 124 (44.0%) cases and increased in 46 (16.3%) cases (*p* < 0.001), while vigorous physical activity was reduced in 110 (39.0%) cases and increased in 27 (9.6%) cases (*p* < 0.001). As suspected, regular physical activity was reduced in FPs during this study period.

### 3.3. The Connection between Burnout and Regular Physical Activity

FPs and FMTs who engaged in regular physical activity did not have lower EE (*p* = 0.105), lower D (*p* = 0.190), higher PA (*p* = 0.067), or lower MBI_TOT_ (*p* = 0.058) scores. All Cohen’s d coefficients resulted in small effect sizes. Contrary to our hypothesis, the incidence of burnout in FPs who were regularly physically active before the COVID-19 pandemic was not lower than in those who were not regularly physically active. (Table 4).

Participants who were regularly physically active during COVID-19 achieved similar scores in EE, D, and PA as those who were not regularly physically active. Overall, high burnout, expressed by MBI_TOT_, was calculated in 43.3% of participants who were regularly physically active and in 52.9% of those who were not regularly physically active. The χ^2^ test showed no statistically significant difference in the proportions according to the levels of burnout (low, medium, high) depending on the practice of regular physical activity during the pandemic (χ^2^ = 2.585; *p* = 0.275). Thus, we were also unable to confirm the hypothesis that the prevalence of burnout was lower in FPs who were regularly physically active during the pandemic than in those who were not regularly physically active.

### 3.4. Occupational Role (FP vs. FMT) and Burnout

FPs reported higher EE and MBI_TOT_ scores compared to FMTs (*p* < 0.001 and *p* = 0.010, respectively). Cohen’s d showed a large effect size for EE and a medium effect size for MBI_TOT_ (Table 5).

## 4. Discussion

The main objective of this study was to determine the prevalence of burnout among FPs and FMTs during the pandemic and to compare it with physical activity and some demographic characteristics.

The most recent data from 2016 on physical activity in Slovenia show that 32.2% of the adult population is not sufficiently physically active and almost 70% of adults are regularly physically active [38]. Studies have shown that physicians are more physically active than the general population, which was also assumed for this study [21]. We found that the majority (62.8%) of physicians were regularly physically active before the pandemic (Table 3), which was lower than data for the adult population in 2016 [38]. Participants may have difficulty recalling physical activities longer than 1 year ago and may have omitted or underreported some leisure-time activities and chores (e.g., walking the dog, mowing the lawn, gardening).

The level of regular physical activity decreased from 62.8% to 45% during the pandemic (FPs and FMTs) (Table 3). This was due to new tasks, additional workload, and more overtime during this period, which likely resulted in increased fatigue. During the pandemic, FPs and FMTs experienced high levels of burnout on each dimension (Table 1 and Table 5); compared with the 2009 FPs survey [39], burnout levels were higher on all three dimensions during the pandemic. The most recent burnout survey of FPs and FMTs in Slovenia in 2017 showed significantly lower burnout levels in all dimensions [40]. The 2017 Slovenian burnout survey showed an improvement of about 25% in EE compared with the 2009 Slovenian survey and also an improvement of about 20% in EE compared with the 2008 EU survey [7,39,40]. This may be because EU initiatives to train more doctors have led to a 13.2% increase in the number of doctors per 100,000 people in the EU between 2004 and 2014. In the decade before that, the increase was only 11.1% [41]. More employment to reduce the burden is an important issue for health professionals, but the pandemic has reversed these recent positive trends, and the latest EU data show a significant shortage of physicians in Slovenia [42].

Compared with a 2008 study from 12 European countries (Slovenia was not included) [7], the present study showed that the pandemic period contributed to higher overall burnout and higher burnout rates in all dimensions. This was observed among physicians working directly with COVID-19 patients [33,34]. Increased job demands during this global health crisis likely explain the decreased leisure time and increase in burnout. At the height of the pandemic in Slovenia, the work style in primary care clinics changed. Fewer patients were physically examined, and many were treated remotely, either by phone or by e-mail. This style of care was more demanding, lacked important information that could have been obtained through clinical examination, and was therefore potentially more burdensome and stressful.

Our results showed no association between overall burnout and regular physical activity (Table 4). Studies in the general population and among physicians worldwide have shown a lower prevalence of burnout among those who are regularly physically active, especially in EE; some other studies have found no significant difference in individual or general burnout [19,21,22,23,24].

The results of this study show that the likelihood of being burned out was significantly higher for FPs than for FMTs (Table 5). This is concordant with findings that residency/clinical fellowship was associated with burnout in internal medicine and primary care physicians [43,44,45]. Other studies have shown opposite patterns [46,47]. Some found that younger physicians were more prone to burnout, whereas others found a negative correlation between age and burnout [2,7,48]. Maslach hypothesized that the reason for higher burnout in younger people is their professional inexperience, so the risk of burnout is greater early in a career. Younger age seems to be associated with professional and personal inexperience, excessive expectations, starting a family, and solving the housing problem [20]. Another explanation for our result could be that FMTs in Slovenia have fewer responsibilities than FPs. FPs were already prone to burnout before COVID-19, which exacerbated preexisting problems when the healthcare system was crippled by the pandemic. FPs are also accustomed to certain work routines that were disrupted during the pandemic. The new and additional healthcare tasks may have contributed to increased stress that led to feelings of burnout [49]. Burnout was particularly pronounced among FPs in the EE dimension (Table 5). During the pandemic, the responsibility of FPs in Slovenia increased because, although they were available for their regular patients, they also had to care for the newly infected COVID-19 patients. They had to adjust to constantly changing guidelines, lack of knowledge, high mortality rates, fear of infection and of infecting family members, personal problems, and, especially for older physicians, personal health problems. All of these factors likely contributed to the higher scores for EE. However, as in other studies [7,39,40], PA was the least-frequently cited value in our study. During the pandemic, regular physical activity decreased by almost 20%, and the amount of weekly physical activity, both moderate and intense, decreased by almost 30%. However, the survey took place during the pandemic and asked about physical activity at two points in time—before and during the pandemic; a pandemic was a specific time period that did not reflect the real picture of a quieter, “normal” time. The self-assessment of physical activity in one week is already somewhat unreliable for the current period/trend. Participants may have made a mistake in estimating the duration of physical activity and have difficulty distinguishing between moderate and vigorous physical activity. Even more difficult and therefore unreliable is the recall of the period before the pandemic, i.e., more than a year before data collection.

Although regular physical activity was not correlated with overall burnout, increased work stress, significant decreases in physical activity, and associated physical decline may have contributed significantly to the findings of EE and PA (Table 5). In previous national and international studies, PA was always less than one-third. We suspect that COVID-19 contributed to this very critical self-assessment of job performance as well as to a general dissatisfaction with life and work circumstances at the time of the survey.

### Study Limitations

The major limitation is the small number of participating FMTs, i.e., less than 20% are registered in the Slovenian Medical Association registry, compared with about 25% of FPs. Because the survey was based on voluntary participation, we could only speculate about the reasons for refusal to participate in this national survey.

A cross-sectional survey design is inherently limited and, together with the fact that we relied on self-reporting, raises the question of the extent to which our results can be explained by methodological variance. However, the phenomenon under study could only be assessed by asking participants to report their experiences or perceptions. Prospective studies with clear criteria and measures, as well as in-depth qualitative studies, would be beneficial to broaden and deepen our understanding of the underlying patterns and associations between physical activity, work status, and burnout. We believe that further research should also focus on a longer time period to provide more detailed characteristics and better bases for planning preventive interventions at the social level and also in education.

## 5. Conclusions

The proportion of FPs and FMTs who engaged in regular physical activity in the period before the pandemic outbreak was lower than expected, and this number continued to decline significantly during the pandemic. Nevertheless, physical activity did not appear to be a statistically significant factor in the occurrence of burnout under the extreme circumstances during the pandemic. Physical activity may nevertheless play a role in coping with stress and burnout, not as a stand-alone activity, but as part of a multidimensional coping approach.

Burnout is a major problem among FPs and FMTs. During the pandemic, high levels of burnout were observed overall and in each dimension. These levels were higher than those observed previously. Burnout overall and EE occurred significantly more frequently among FPs. During the first height of the epidemic in Slovenia, the way of working in the FP clinics changed, as the already too few FPs were exposed to a high workload and responsibility. They were exposed to a high risk of COVID-19 infection, which probably led to concerns about their own health and that of their families. The latter should be investigated in further studies. Therefore, working conditions, work organization, and workload should take priority over physical activity in burnout prevention.

## Figures and Tables

**Table 1 healthcare-12-00028-t001:** Demographic characteristics and burnout according to occupational role (FMs vs. FMTs).

Demographics	Total n = 282 (%)	Work Status/Occupational Role		*p*
		Family Medicine Trainees (FMTs) n = 39 (%)	Family Physicians (FPs) n = 243 (%)	
**Gender**				0.145 *
Female	221 (78.4)	27 (69.2)	194 (79.8)	
Male	61 (21.6)	12 (30.8)	49 (20.2)	
**Age in years**				<0.001 #
Below 30	15 (5.3)	15 (38.5)	0 (0.0)	
30–39	98 (34.8)	23 (59.0)	75 (30.9)	
40–49	79 (28.0)	1 (2.6)	78 (32.1)	
50–59	57 (20.2)	0 (0.0)	57 (23.5)	
60 or above	33 (11.7)	0 (0.0)	33 (13.6)	
**MBI**				0.022 #
High	137 (48.6)	11 (28.2)	126 (51.9)	
Medium	90 (31.9)	18 (46.2)	72 (29.6)	
Low	55 (19.5)	10 (25.6)	45 (18.5)	

* Fisher’s exact test, # chi-square test.

**Table 2 healthcare-12-00028-t002:** The prevalence of emotional exhaustion, depersonalization, personal overload, and general burnout among participants during the pandemic.

Rate	Emotional Exhaustion (EE) n (%)	Depersonalization (D) n (%)	Personal Accomplishment (PA) n (%)	Total Burnout Score (MBI_TOT)_) n (%)
High	177 (62.8)	113 (40.1)	30 (10.6)	137 (48.6)
Medium	59 (20.9)	108 (38.3)	101 (35.8)	90 (31.9)
Low	46 (16.3)	61 (21.6)	151 (53.5)	55 (19.5)

**Table 3 healthcare-12-00028-t003:** Moderate and vigorous physical activity before and during the COVID-19.

Weekly Duration of Physical Activity (in min)	Moderate Physical Activity		Weekly Duration of Physical Activity (in min)	Vigorous Physical Activity	
	Before COVID-19 n = 282 (%)	During COVID-19 n = 282 (%)		Before COVID-19 n = 282 (%)	During COVID-19 n = 282 (%)
Less than 30	24 (8.5)	63 (22.3)	Less than 15	83 (29.4)	124 (44.0)
30–59	46 (16.3)	54 (19.1)	15–29	48 (17.0)	53 (18.8)
60–119	53 (18.8)	49 (17.4)	30–59	60 (21.3)	46 (16.3)
120–149	59 (20.9)	39 (13.8)	60–89	29 (10.3)	18 (6.4)
150–209	44 (15.6)	31 (11.0)	90–119	28 (9.9)	21 (7.4)
210–299	27 (9.6)	27 (9.6)	120–149	22 (7.8)	12 (4.3)
300 or more	29 (10.3)	19 (6.7)	150 or more	12 (4.3)	8 (2.8)

**Table 4 healthcare-12-00028-t004:** Signs of emotional exhaustion, depersonalization, personal accomplishment, and total burnout in relation to regular physical activity during the pandemic.

MBI Dimension	Regular Physical Activity		Cohen’s d	*t*	*p* *
	YES n = 127; M ± SD	NO n = 155; M ± SD			
Emotional exhaustion (EE)	29.1 ± 11.9	31.5 ± 13.2	0.19	1.627	0.105
Depersonalization (D)	10.6 ± 5.2	11.5 ± 5.9	0.16	1.314	0.190
Personal accomplishment (PA)	31.7 ± 6.0	30.3 ± 6.0	0.25	1.841	0.067
Total burnout score MBI_TOT_	55.9 ± 20.9	60.7 ± 21.8	0.22	1.903	0.058

* independent-samples *t*-test.

**Table 5 healthcare-12-00028-t005:** Signs of emotional exhaustion, depersonalization, personal accomplishment, and overall burnout according to work status/occupational role.

MBI Dimension	Work Status/Occupational Role		Cohen’s d	*t*	*p* *
	Family Medicine Trainees (FMTs) n = 39; M ± SD	Family Physicians (FPs) n = 243; M ± SD			
Emotional exhaustion (EE)	22.6 ± 10.2	31.7 ± 12.6	0.79	4.263	<0.001
Depersonalization (D)	10.2 ± 5.5	11.2 ± 5.6	0.18	1.095	0.275
Personal accomplishment (PA)	30.5 ± 5.9	31.0 ± 6.1	0.08	0.538	0.591
Total burnout score MBI_TOT_	50.3 ± 19.5	59.9 ± 21.6	0.47	2.599	0.010

* independent-samples *t*-test.

## Data Availability

Data sets used and/or analyzed in the current study are available upon request from the corresponding author.

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
