# Peer review of "A Cross-Sectional Study on Physical Activity and Burnout among Family Physicians in Slovenia during the First Year of the COVID-19 Pandemic: Are the Results Alarming Enough to Convince Decision-Makers to Support Family Medicine?"

_healthcare, 2023, doi:10.3390/healthcare12010028_

Round 1

Reviewer 1 Report

Comments and Suggestions for Authors

The article deals with an important topic. The substantive description of the research premises is very valuable.

However, the Authors should enrich the literature review. Currently, a lot of articles have been published on Covid-19 and professional burnout. These articles deal with the theory of the psychology of occupational burnout and its relevance to occupational burnout in the era of Covid-19. Many studies are based on primary research conducted among professional and socio-demographic groups examining their responses to the pandemic and lockdown. The Authors of the reviewed article limited themselves to only a few items. For example, the Authors can use:

- Heavner, S.F.; Stuenkel, M.; Russ Sellers, R.; McCallus, R.; Dean, K.D.; Wilson, C.; Shuffler, M.; Britt, T.W.; Stark Taylor, S.; Benedum, M.; et al. “I Don’t Want to Go to Work”: A Mixed-Methods Analysis of Healthcare Worker Experiences from the Front- and Side-Lines of COVID-19. Int. J. Environ. Res. Public Health 2023, 20, 5953. https://doi.org/10.3390/ijerph20115953

- Kwiatkowska-Ciotucha, D., Załuska, U., Ślazyk-Sobol, M., Lehesvuori, M. & Polak, A. (2019). Occupational Burnout in Health Care – Analysis of Systemic and Organisational Risks as Well as Possible Preventive Actions. Econometrics, 23(4) 43-62. https://doi.org/10.15611/eada.2019.4.04

- Zaluska, U., Kwiatkowska-Ciotucha, D. & Slazyk-Sobol, M. (2020). Burnout Syndrome as the Example of Psychological Costs of Work – Empirical Studies among Human-Oriented Professions in Poland. IBIMA Business Review, Vol. 2020 (2020), Article ID 430264, DOI: 10.5171/2020.430264

- Załuska,U., Ślazyk-Sobol, M. & Kwiatkowska-Ciotucha, D. (2018). Burnout and Its Correlates − An Empirical Study Conducted Among Education, Higher Education and Health Care Professionals. Econometrics, 22(1) 26-38. https://doi.org/10.15611/eada.2018.1.02

The Introduction is essentially a literature review. I suggest that the authors rewrite the Introduction, explaining the research problem and the purpose of the study. It is crucial that they also state what hypotheses or research questions they pose. In the Introduction, they should also briefly describe what is in the subsequent chapters of the work.

The results in Table 2 are not discussed in the text immediately below the table. Only in lines 169-173 are there references to tables 2 and 3. Additionally, considering the key indicators of the analysis, as the authors wrote in line 127, the conclusions from these two tables are very scanty.

The abbreviations are explained the first time you use them, so there is no need to explain them again at the end of the article.

Author Response

All authors are grateful for the comments and have done their best to incorporate and summarize them in the improved and revised version of the manuscript. The red color of the fonts has been used for added or changed text.

Reviewer 2 Report

Comments and Suggestions for Authors

The manuscript called "Physical activity and burnout among family physicians in Slovenia during the COVID -19 pandemic: are the results alarming enough to prompt decision makers to support family medicine?" is about a very important them. Burnout is a concept which is involved in multiples scenarios and professional collectives. Therefore, it is necessary to keep on studying this issue. With regard to considerations to improve the manuscript:

1. It should be necessary to explicit the specific goals developed from the main goal. 

2. To structure the section "Results" depending on the specific goals previouly exposed in the introduction.

3. To explicit a subsection called "Procedure" within Method and Materials. Not only participants, measures, and data analysis.

4. Describe a little more the instruments, by including an item for each dimension of the questionnaire. 

Thank you very much for your attention.

Author Response

(The authors gave the same response as above.)

Reviewer 3 Report

Comments and Suggestions for Authors

Dear author, 

First I would like to congratulate you for choosing this important topic for your research. On the whole, the article makes a very good impression. I have a few suggestions which could improve your manuscript.

Title

During COVID-19 is a long period...Maybe the authors could change the Title to be more precise about the time of the study, and also about the methodology (A cross-sectional study).

Abstract

The abstract is good, but the last sentence needs to be changed. Work status and maybe other factors should be investigated in future studies... this study didn't confirm it is a risk factor.

Introduction

The Introduction is very well written... Paragraph 1 line 39 (and is common in family medicine), please cite this conclusion.

Methodology

It is described well and clearly.

Result

You could add Table demographic characteristics in the total sample and also stratified by the workplace (FMTs, FPs). I think it would be clearer and easier to read for readers.

Discussion

The author discusses their result and compares them with other studies. Also, they add limitations to the study.

Conclusion

The last sentence is just like in the Abstract. Please change it to be in accordance with this study.

Author Response

(The authors gave the same response as above.)
